# Extramedullary Relapse of *CBFA2T3::GLIS2*-Positive Megakaryoblastic Leukemia Mimicking Secondary Ewing Sarcoma: An Exemplary Case for the Diagnostic Trap

**DOI:** 10.3390/ijms26125895

**Published:** 2025-06-19

**Authors:** Svetlana Lebedeva, Ekaterina Mikhailova, Sophia Bogacheva, Dmitry Abramov, Svetlana Kashpor, Alexander Druy, Alexandra Semchenkova, Marina Gaskova, Olga Lotonina, Ilya Sidorov, Galina Tereschenko, Yulia Olshanskaya, Galina Novichkova, Alexey Maschan, Elena Zerkalenkova, Alexander Popov

**Affiliations:** Dmitry Rogachev National Medical Research Center of Pediatric Hematology, Oncology and Immunology, 1, S. Mashela St., Moscow 117998, Russia; katmikhailova1805@gmail.com (E.M.); sofya.bogacheva@dgoi.ru (S.B.); abramovd_s@bk.ru (D.A.); svetlana.kashpor@dgoi.ru (S.K.); dr-drui@ya.ru (A.D.); marina.gaskova@dgoi.ru (M.G.); olga.lotonina@dgoi.ru (O.L.); ilya.sidorov@dgoi.ru (I.S.); galina.tereschenko@dgoi.ru (G.T.); yuliya.olshanskaya@dgoi.ru (Y.O.); gnovichkova@yandex.ru (G.N.); amaschan@mail.ru (A.M.); uralcytometry@gmail.com (A.P.)

**Keywords:** acute megakaryoblastic leukemia, inv(16)(p13q24)/*CBFA2T3::GLIS2*, extramedullary relapse

## Abstract

In children without Down syndrome who have acute megakaryoblastic leukemia (AMKL), inv(16)(p13q24)/*CBFA2T3::GLIS2* is the most frequent genetic aberration. Pediatric *CBFA2T3::GLIS2*-positive AMKL is strongly associated with a poor prognosis and a high cumulative incidence of relapse. One of the key laboratory signs of *CBFA2T3::GLIS2*-positive AMKL is the RAM immunophenotype, which looks very similar to that of solid-tumor bone marrow (BM) infiltration. For this reason, in cases of isolated extramedullary involvement of *CBFA2T3::GLIS2*-positive AMKL, excluding solid tumors may be challenging. We report a case of a girl with isolated extramedullary *CBFA2T3::GLIS2*-positive AMKL relapse, which was misdiagnosed as secondary Ewing sarcoma. The morphological differential diagnosis between Ewing sarcoma and AMKL presented significant challenges owing to their overlapping histological features (small, round blue-cell morphology and similar growth patterns). The tumor cells’ immunophenotype completely mirrored that at the initial diagnosis of AMKL. Additional cytogenetic and molecular studies confirmed the presence of the *CBFA2T3::GLIS2* fusion, but no Ewing sarcoma-specific *EWSR1, FUS* and *CIC* fusion transcripts were found. Thus, extramedullary *CBFA2T3::GLIS2*-positive AMKL relapse was confirmed. The presented case demonstrates the difficulties in differential diagnosis between AMKL relapse and the development of a secondary tumor.

## 1. Introduction

Acute megakaryoblastic leukemia (AMKL) is a rare, molecularly heterogeneous subtype of acute myeloid leukemia (AML) that is usually described as the M7 type of AML according to the FAB classification [1,2,3,4]. The incidence of AMKL ranges from 4 to 15% among pediatric AML cases, and AMKL accounts for approximately 1% of adult AML cases. AMKL can be divided into Down syndrome AMKL (DS-AMKL) and non-DS-AMKL, which is associated with a worse survival prognosis [5,6,7,8,9]. *CBFA2T3::GLIS2*, resulting from cryptic inversion of chromosome 16, is the most frequent chimeric oncogene in non-Down syndrome AMKL [1,10,11,12,13]. The presence of this fusion gene has been described predominantly in infants and young children, and it is associated with an extremely poor prognosis (a 5-year overall survival rate of less than 34.3%) and a high cumulative incidence of relapse [10,11,14,15]. One of the key laboratory signs of *CBFA2T3::GLIS2*-positive AMKL is the so-called RAM immunophenotype [16,17,18], which is characterized by dim-to-negative CD45 and CD38 expression, extremely bright CD56 and a lack of HLA-DR on leukemic cells [17,19]. This antigen expression profile looks very similar to that of solid-tumor bone marrow infiltration [17,20,21], whereas in *CBFA2T3::GLIS2*-positive AMKL, the expression of typical myeloid and progenitor markers (CD33, CD117 and CD34) is usually visible [17,18,21,22]. Moreover, the expression of platelet markers (CD61, CD41a and CD42b) is always indicative of AMKL [23,24].

Pediatric AML often manifests with extramedullary lesions, including myeloid sarcoma (MS), also known as chloroma [25,26,27,28], which is a solid lesion of extramedullary accumulation of malignant myeloid cells or their precursors [4]. Establishing the diagnosis of AMKL in cases of isolated extensive extramedullary involvement may be challenging. In addition to the difficulties that arise in differential diagnosis using radiation methods, pathological diagnostics can also produce a contradictory picture when identifying and classifying atypical myeloid cells [29,30].

Herein, we present a case of isolated extramedullary relapse of *CBFA2T3::GLIS2*-positive AMKL after a second allogeneic hematopoietic stem cell transplantation (allo-HSCT) in a 3.5-year-old girl who was misdiagnosed with secondary Ewing sarcoma.

## 2. Case Presentation

A 1.7-year-old girl was referred to a local hospital because of fever, pallor and cytopenia. Laboratory tests revealed bicytopenia (Hb 74 g/L, total WBC count 5.4 × 10^9^/L, neutrophils 2.7 × 10^9^/L and platelet count 580 × 10^9^/L) and an increased LDH level (754 units/L). An abdominal ultrasound revealed mild hepatosplenomegaly. Morphological evaluation of the bone marrow aspirate revealed 96% blasts with megakaryocytic morphology (Figure 1A). Bone marrow immunophenotyping revealed 95% CD45-negative blasts with bright expression of CD33, CD117 and CD56 and characteristic features of AMKL, including CD11a negativity and expression of the platelet antigens CD61 and CD41a (Figure 2A). Conventional cytogenetic evaluation of the bone marrow aspirate revealed isolated trisomy 3, and inv(16)(p13q24)/*CBFA2T3::GLIS2* was found in 90% of the nuclei by FISH (Figure 1C). RNA-seq of the BM aspirate revealed *CBFA2T3::GLIS2* fusion transcript expression with an exon 9–exon 3 junction. No accompanying mutations in AML-related genes were detected by NGS. No leukemic cells were detected in the cerebrospinal fluid.

Thus, the diagnosis of AMKL with inv(16)(p13q24)/*CBFA2T3::GLIS2* was established, and the patient received AML-MRD-2018 protocol induction therapy, consisting of cytarabine, etoposide and idarubicin [31,32]. According to the protocol’s stratification system [31], the patient was allocated to the high-risk (HR) group. Although complete remission (CR) was achieved after induction, a high level of Minimal Residual Disease measured by Multifactorial Flow Cytometry (MFC-MRD) (1.541%) was detected (Figure 2B). In addition, RT–PCR negativity was not reached, with only a 2-log reduction in relative *CBFA2T3::GLIS2* expression. The treatment continued with high-dose cytarabine in conjunction with fludarabine and idarubicin (FLAI course), while the patient remained highly MFC-MRD refractory (1.172%), and after the third cycle of chemotherapy with HDaraC and fludarabine (FLA), the MFC-MRD level even increased to 2.122% (Figure 2B). Thus, a decision was made to proceed to hematopoietic stem cell transplantation (HSCT) without further attempts to achieve MRD-negative remission. After myeloablative conditioning with treosulfan 42 g/m^2^, fludarabine and thiotepa 10 mg/kg b.w., the patient received HSCT from her HLA-identical sibling. One month after HSCT, MFC-MRD (0.013%) was still detectable in the BM, and *CBFA2T3::GLIS2* expression was also positive. To prevent overt leukemia relapse, the patient was treated with venetoclax and decitabine. Eight months after the initial diagnosis and 3 months after HSCT, the MFC-MRD level increased to 2.685%. Low-dose cytarabine in combination with 6-mercaptopurine was administered, and donor lymphocyte infusions were given, with no influence on the MRD level, which remained high at 1.184%. Owing to inevitable AMKL relapse, a second haploidentical HSCT was performed 10 months after diagnosis (5 months after HSCT #1). On day +31 after the second HSCT, MFC-MRD-negative and RT–PCR-negative remission was documented for the first time.

On day +143 after the second HSCT, MFC-MRD reappeared (0.028%; Figure 2B). The patient was treated with azacitidine and bortezomib, resulting in subsequent MFC-MRD elimination on day +165. Three months later, at the age of 3,5 – years, the patient was admitted with pain in the right hip area.

To determine the cause and localization of pain in the right hip, with subsequent surgical planning of biopsy, a traditional computed tomography (CT) scan and diffusion-weighted (DW) magnetic resonance imaging (MRI) were performed. The imaging data revealed a periosteal reaction of the proximal metadiaphysis of the right femur, an extraosseous soft-tissue mass up to 4 cm in length and focal liver lesions. The extramedullary component semicircularly surrounded the metadiaphysis of the femur, and was characterized by an isointense signal on the T1-weighted image and a hyperintense signal on the T2-weighted image relative to muscle tissue; there was an absence of contrast-agent accumulation, necrotic cavities and hemorrhage areas in the structure (Figure 3). MRI data showed high signal intensity on DWI and decreased signal intensity on ADC maps, indicating the restriction of water molecule diffusion due to the high cellularity and malignancy of the tumor (Figure 3A(e,f)) [33]. When selecting regions of interest (ROIs) in the extramedullary mass structure for ADC value measurements, MR scans were compared with CT images to exclude bone density areas. The obtained ADC values of the extramedullary mass ranged from 0.240 × 10^−3^ to 0.426 × 10^−3^ mm^2^/s, which were lower than previously reported values in Ewing sarcoma cases [34,35].

The common localization in the proximal metadiaphysis of the femur, with an extramedullary component and intramedullary infiltrative changes in the bone marrow, mimicked the early development of the bone form of Ewing’s sarcoma (Figure 3B) [36,37,38]. However, the avascular structure of the lesion, low ADC values and clinical symptoms were more characteristic of hematological malignancy [29,39].

The MFC-MRD results were negative in the BM. On the basis of the MFC-MRD results and imaging data, a secondary bone tumor was suspected, rather than leukemia relapse. A core biopsy of the right femur was performed, but the material obtained was insufficient for reliably verifying the diagnosis. On the basis of the first histological examination result, a diagnosis of Ewing sarcoma was suspected.

## 3. Laboratory Studies at the Time of Suspected Relapse Diagnosis

The patient was admitted to Dmitry Rogachev National Medical Research Center of Pediatric Hematology, Oncology and Immunology to repeat biopsy and verify the diagnosis. An open biopsy of the right femur and extraosseous soft-tissue component was performed. The tumor demonstrated a distinctive morphological pattern, characterized by sheets and nests of monomorphic small, round cells infiltrating the intertrabecular spaces. The neoplastic cells presented high nuclear-to-cytoplasmic ratios, with round-to-oval nuclei containing finely dispersed chromatin and inconspicuous nucleoli. Cellular proliferation was intersected by prominent fibrous trabeculae, creating a compartmentalized architecture. Focal areas of geographic necrosis and hemorrhage were present throughout the lesion. The tumor cells displayed brisk mitotic activity with numerous atypical mitotic figures. The overall histological appearance bore a striking resemblance to Ewing sarcoma, necessitating careful immunohistochemical evaluation for definitive diagnosis (Figure 4). The expression of FLI-1, ERG, and LMO2 was detected. At the same time, tumor cells were negative for MPO, CD99, NKX2.2 and CD61. The tumor material was subjected to molecular testing to detect genetic aberrations specific for Ewing sarcoma. However, preliminary analysis of the extracted RNA from the obtained biopsy material revealed a high level of fragmentation (median fragment lengths < 90 bases), making it impossible to study fusion transcripts by reverse-transcriptase PCR or RNA-seq. Considering the patient’s initial diagnosis of AMKL, controversial cell morphology, and similarity to the immunophenotypes of Ewing sarcoma and *CBFA2T3::GLIS2*-positive AML, as well as the lack of molecular diagnostics, comprehensive additional immunophenotypic and molecular studies were performed to confirm the diagnosis of a secondary tumor, rather than extramedullary recurrence of AML.

BM microscopy revealed the absence of any signs of tumor lesions in three of the four puncture sites; although in one specimen, a conglomerate of suspicious cells was found, confirming the tumor nature of these cells was impossible (Figure 1B). MFC-MRD was detected in 0.028% of residual leukemic cells, which were CD45-negative, CD33-positive, CD117-positive, CD11a-negative and HLA-DR-negative, and also highly expressed CD56 and were heterogeneously positive for CD34. This immunophenotype completely mirrored that at the initial diagnosis of AMKL. The cells were isolated by flow cell sorting, and FISH confirmed the presence of the *CBFA2T3::GLIS2* gene fusion (Figure 1D). RT–PCR-based investigation of the whole BM sample revealed the presence of the respective fusion transcript (a 1-log reduction in *CBFA2T3::GLIS2* expression from the time of diagnosis). Considering that AMKL cells had reappeared in the BM, a new biopsy of the tumor mass in the femur was performed. An MFC study of the tumor mass revealed cells with an immunophenotype that completely resembled that of the initially diagnosed AMKL, which was confirmed by positivity for the thrombocytic antigens CD61 and CD41a (Figure 2C). A FISH study of intraosseous tumor dabs revealed *CBFA2T3::GLIS2* fusion formation and the absence of *EWSR1* rearrangement (Figure 1E,F). A pure tumor-cell suspension was isolated from the homogenized biopsy material by flow cell sorting for additional molecular studies. FISH of the sorted cells confirmed the presence of the *CBFA2T3::GLIS2* fusion. RNA-seq analysis of the tumor biopsy material confirmed the expression of the *CBFA2T3::GLIS2* fusion transcript, with its sequence fully matching that obtained in the initial RNA-seq results (Figure 1G). No Ewing sarcoma-specific *EWSR1*, *FUS* and *CIC* fusion transcripts were found.

Finally, the diagnosis was changed from secondary Ewing sarcoma to extramedullary relapse of AMKL with inv(16)(p13q24)/*CBFA2T3::GLIS2*. Palliative chemotherapy with low-dose cytarabine in conjunction with 6-mercaptopurine was administered. The patient’s clinical condition gradually worsened, and the patient died 5 months after biopsy and verification of extramedullary relapse of AMKL.

## 4. Discussion

AMKL is a very heterogeneous form of AML [23,40], and is defined by the presence of leukemic megakaryoblastic cells expressing platelet-specific surface glycoproteins [24,41]. AMKL is significantly more common in children than in adults, accounting for 4–15% of pediatric AML cases, as reported by large collaborative studies [11,12,13]. Except for the expression of CD61, CD41 and isoforms of CD42, the antigen expression profiles of AMKL cells vary greatly [24], in accordance with the genetic subgroups of this AML type. In non-Down syndrome children with AMKL, *CBFA2T3::GLIS2* is the most frequent chimeric oncogene identified to date [1,11,42]. The most common chimeric *CBFA2T3::GLIS2* transcript is formed between exon 11 of *CBFA2T3* and exon 3 of *GLIS2* [11,15,43], but other rare fusion transcripts, such as *CBFA2T3*-ex10/*GLIS2*-ex3, *CBFA2T3*-ex12/*GLIS2*-ex1 [11] and *CBFA2T3*-ex10/*GLIS2*-ex2 [15], have also been reported. AMKL with *CBFA2T3::GLIS2* has a very specific immunophenotype, which is usually referred to as the RAM phenotype [16,17,18]. Although the majority of AMKL cases are CD45-positive [24,40], the RAM phenotype belongs to the CD45-negative AMKL subgroup. Moreover, these patients are characterized by extremely bright CD56 and the absence of HLA-DR and CD38 [17,19]. Furthermore, CD56+CD45−-cells are always suggestive of BM metastases of various solid tumors [17,20,21,44], and even BM differentiation between leukemia and nonhematopoietic tumors is one of the main challenges in *CBFA2T3::GLIS2*-positive AMKL diagnosis [17,21]. Pediatric AMKL with the *CBFA2T3::GLIS2* fusion is strongly associated with a poor prognosis, because of the high frequency of primary induction failure and a high cumulative incidence of relapse [1,15]. Considering the very poor treatment response and initial BM relapse after the first HSCT in the present case, the second relapse was absolutely expectable. Moreover, extramedullary involvement both at diagnosis and in relapse is relatively frequent in *CBFA2T3::GLIS2*-positive leukemia patients compared with AML patients without this genetic aberration [15,45,46].

Ewing sarcoma is diagnosed very rarely as a secondary malignant neoplasm [47,48,49,50], accounting for less than 1.5% of all secondary tumors after the treatment of all childhood cancers [47,51]. In our case, CT and MRI revealed a characteristic localization of the tumor in the proximal metadiaphysis of the femur, with cortical destruction and a spiculated periosteal reaction, which could indeed be interpreted as Ewing sarcoma [52].

Myeloid sarcoma is radiologically non-specific and can occur in any area of the human body. However, if there is a history of leukemia, any extramedullar lesion is suspected of being a relapse [52]. Our case demonstrates an unusual location for AMKL extramedullary relapse with the cortical bone destruction typical in Ewing’s sarcoma.

In studies of musculoskeletal tumors, the approximate mean ADC is possible to differentiate between early and malignant bone tumors: an ADC value < 1.03 × 10^−3^ mm^2^/s in the pediatric group [35] and an ADC value of 1–1.31 × 10^−3^ mm^2^/s for children and young adults [53]. The ADC values for different malignancies are variable; according to recent studies, Ewing’s sarcoma has the lowest mean ADC, from 0.5 × 10^−3^ to 0.8 × 10^−3^ mm^2^/s [34,35,53,54]. The ADC values obtained for the extramedullary mass in our case ranged from 0.240 × 10^−3^ to 0.426 × 10^−3^ mm^2^/s, consistent with one of the few studies of bone marrow infiltration in children leukemia and lymphoma conducted by Wang, Haoyu et al. [39]. Also, isolated cases of myeloid sarcomas with available diffusion-weighted MRI data where the minimum ADC value was 0.36 × 10^−3^ mm^2^/s have been described previously [29,55].

While DWI and ADC maps offer crucial insights into imaging data, the absence of universal cut-offs and established thresholds complicates their use in definitive diagnostics. These limitations render them supplementary tools rather than standalone methods, necessitating histological validation for accurate tumor characterization.

The differential diagnosis between leukemia relapse and secondary tumors is crucial in the case of secondary leukemia development. It is typically necessary to prove the diagnosis of secondary tumors, rather than a lineage switch at relapse, for types of leukemia that demonstrate increased lineage plasticity [56,57,58], such as AL with *KMT2A*, *ZNF384* or *DUX4* gene rearrangements or early T-cell acute lymphoblastic leukemia (ETP-ALL) [59,60]. For such cases, the main tumor site is the BM, and the differences in morphology and/or immunophenotype between initially diagnosed leukemia and secondary leukemia make it necessary to confirm relapse or a secondary tumor by molecular studies [60]. In our patient, we observed no differences in immunophenotype between initial AMKL and BM leukemic cells at the MFC-MRD level or extramedullary relapse. Nevertheless, the specific location of the small-, round-cell tumor without BM involvement, even at the MFC-MRD level, led to the misinterpretation of relapse development as secondary Ewing sarcoma. Typically, distinguishing between solid tumors and acute leukemia by antigen expression profile analysis is not difficult [20], as the vast majority of AL patients express hematopoietic markers, including the panleukocytic antigen CD45 and markers of either lymphoid or myeloid lineages [61], whereas the immunophenotypes of nonhematopoietic tumors are significantly different, and CD45 negativity is the obligatory feature of nonhematopoietic cells [20]. However, in the presented case, these differences were not as obvious. As mentioned above, excluding solid tumors is always one of the main questions in the diagnosis of AMKL with the RAM phenotype [17,19]. Although it is faster and easier to perform using data from the BM [17], differential diagnosis of isolated extramedullary tumors seems much more difficult. The morphological differential diagnosis between Ewing sarcoma and megakaryoblastic sarcoma presents significant challenges owing to their overlapping histological features, including small, round blue-cell morphology and similar growth patterns. Immunohistochemically, both tumors can express CD99, which is traditionally considered a marker for Ewing sarcoma; however, megakaryoblastic sarcoma may also show variable expression of this marker, making its interpretation more complex [62]. A definitive diagnosis often requires a comprehensive panel of immunohistochemical markers, including specific megakaryocytic markers (such as CD41, CD61, and Factor VIII), CD99, FLI1 and NKX2.2), along with molecular testing for EWSR1 rearrangement, as no single marker is exclusively specific for either entity, and the interpretation must be performed in the context of the complete clinical and pathological picture [63]. In the presented case, the immunohistochemistry results were completely controversial. On the one hand, CD99 negativity is not typical for Ewing sarcoma. On the other hand, FLI-1 expression is more characteristic of Ewing sarcoma, while the absence of CD61 in immunohistochemical studies also questions the diagnosis of AMKL. In such complicated cases, only MFC, with its wide range of applicable antigens, and comprehensive genetic studies are always capable of confirming the development of MS or nonhematopoietic tumors. Although MFC can provide clinically relevant data more quickly, all such cases of supposed secondary tumors should undergo extensive genetic studies, because the final confirmation of the presence and type of solid tumor will be based on molecular findings.

In summary, the presented case demonstrates the difficulties of differential diagnosis between leukemia relapse and the development of a secondary tumor. Not only significant changes in the presenting features, but also atypical locations, can lead to misdiagnosis and incorrect treatment. While making a diagnosis solely based on one method could be insufficient, in-depth laboratory and radiology studies, together with accurate and precise anamnesis investigations, can be effective enough for defining the correct tumor type and subsequently selecting an appropriate treatment strategy.

## 5. Methods

### 5.1. Immunophenotyping

Analysis of the immunophenotype of tumor cells was performed using 10-color combinations of monoclonal antibodies (Table 1) according to the diagnostic standards of the Moscow–Berlin group, as described previously [61].

Tubes #1 and #3–6 were custom-designed pre-manufactured tubes (DuraClone™ technology, Beckman Coulter [BC], Brea, CA, USA). Tube #1 was an orientation tube for initial BM investigation. Tube #2 included cytoplasmic antigens for the complete lineage definition. Tubes #3–4 contained typical antibodies for AML diagnostics, while tubes #5–6 were MFC-MRD tubes [64,65], and they were used during the initial diagnostics in order to provide so-called “Day 0” information. In tube #6, PE- and APC-labeled markers were drop-in antibodies, and they were selected according to the observed immunophenotypic aberrations (Table 1 shows the combination with antibodies used in the described case). All the samples were processed according to the manufacturer’s recommendations. At least 30,000 nucleated cells were collected from initial and relapse diagnostic samples by using a Navios (BC, Indianapolis, IN, USA) flow cytometer. The EuroFlow guidelines for machine performance monitoring were used [66]. Flow-Check Pro Fluorospheres (BC) were used for daily cytometer optimization. The results were analyzed with Kaluza Analysis 2.1 (BC). Leukemic cells were gated on the basis of CD45 and lineage-associated antigen expression and side-scatter light characteristics [61]. Residual normal cells preserved in a sample were used as an internal control. MFC-MRD was evaluated according to the FlowCluster group approach [64,65], with an adaptation for the RAM phenotype [17,67]. The antibody combinations for tubes #5–6 from Table 1 were used for the MFC-MRD studies.

Leukemic blasts were isolated for cytogenetic and molecular studies from BM samples and soft-tissue biopsy samples using an FACS Aria III sorter (Becton Dickinson [BD], San Jose, CA, USA), as described previously [68]. Sample preparation depended on the downstream molecular study. For FISH, an erythrocyte lysis buffer with a fixative (FACS Lyse; BD) was used, and the presorted samples were diluted with RPMI-1640 medium (PanEco, Moscow, Russia). For PCR and sequencing, a nonfixative lysis agent (PharmLyse; BD) was used, and the cells were diluted with phosphate-buffered saline (Cell Wash; BD). The cells were sorted in purity mode and collected into Eppendorf tubes containing the relevant buffer. Between 10,000 and 15,000 cells were sorted in duplicate for FISH, and 3 to 5 million cells were sorted in duplicate for molecular studies.

### 5.2. Cytogenetic and Molecular Studies

Conventional GTG-banded karyotyping was performed on bone marrow aspirates, which were cultured according to a previously described method [69]. The results were interpreted following the International System for Human Cytogenomic Nomenclature-2024 [70].

Fluorescence in situ hybridization (FISH) was conducted with the Cytocell CBFA2T3::GLIS2 dual-color dual-fusion probe (Cytocell, Cambridge, UK) and the Kreatech EWSR1 dual-color break-apart probe (Leica Biosystems, Nussloch, Germany). Reverse-transcription–polymerase chain reaction (RT–PCR) for the *CBFA2T3::GLIS2* fusion transcript was performed with the following primers and probes: CBFA2T3-F, ACGCCGAGGACACAAAGAAGGG; GLIS2-R, ACACCATCCAAATAGCGCAG; and GLIS2-P, FAM-CGAGGCACTGGGCACTGGAGGC-BHQ1. Whole-transcriptome sequencing (RNA-seq) was carried out by using the NEBNext Ultra II Directional RNA library preparation Kit (NEB, Ipswich, MA, USA) for bone marrow aspirates, and the VAHTS Universal V8 RNA-seq Kit (Vazyme, Nanjing, China) for a biopsy sample of the extramedullary mass. The samples were sequenced on an Illumina NextSeq platform (Illumina, San Diego, CA, USA) and analyzed using the Arriba fusion detection algorithm [71]. Additionally, mutations in genes relevant to myeloid neoplasm development were examined using the Qiagen human Myeloid Neoplasms panel (Qiagen, Hilden, Germany). Sequencing was performed on an Illumina MiSeq platform (Illumina, San Diego, CA, USA), and data analysis was conducted using the Qiagen built-in pipeline.

## Figures and Tables

**Figure 1 ijms-26-05895-f001:**
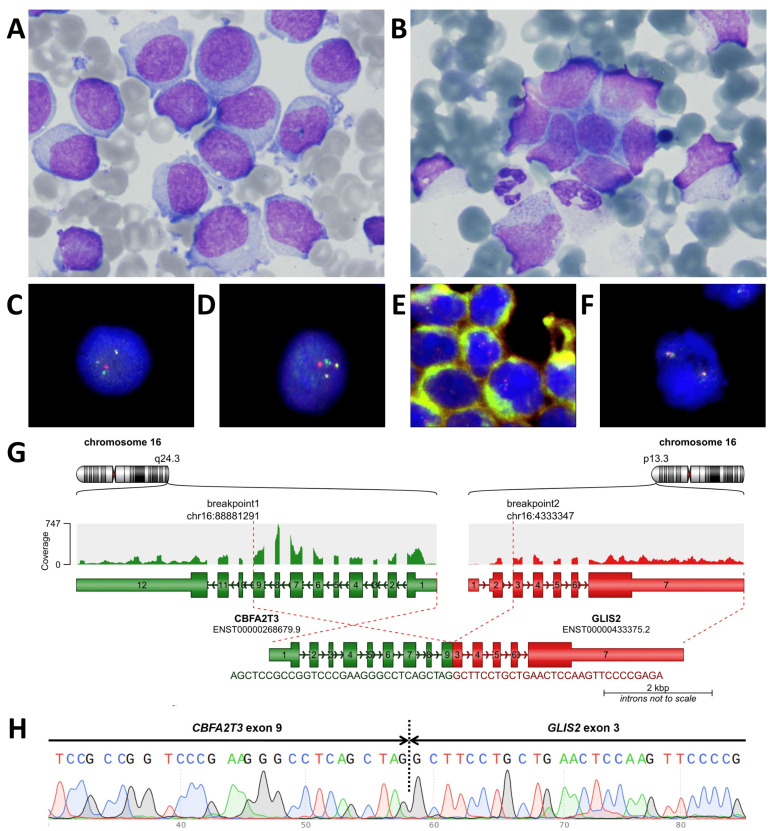
Morphological, cytogenetic and molecular examination of primary *CBFA2T3::GLIS2*-positive AMKL and isolated extramedullary relapse samples. (**A**)—The morphology of the BM in primary AMKL; (**B**)—the morphology of a BM cell conglomerate in a relapse sample; (**C**)—FISH with the Cytocell CBFA2T3::GLIS2 dual-color dual-fusion probe (Cytocell, Cambridge, UK) in the BM in primary AMKL, *GLIS2* is marked with green fluorophore, *CBFA2T3* is marked with orange fluorophore; (**D**)—FISH with the CBFA2T3::GLIS2 probe in the flow-sorted BM population from a second relapse sample; (**E**)—FISH with the CBFA2T3::GLIS2 probe in the intraosseous tumor dab from relapse; (**F**)—FISH with the Kreatech EWSR1 dual-color break-apart probe (Leica Biosystems, Nussloch, Germany) in the intraosseous tumor dab from relapse; (**G**)—the *CBFA2T3::GLIS2* fusion transcript detected by RNA-seq in a relapse sample; (**H**)—the *CBFA2T3::GLIS2* fusion transcript confirmed by Sanger sequencing in a relapse sample.

**Figure 2 ijms-26-05895-f002:**
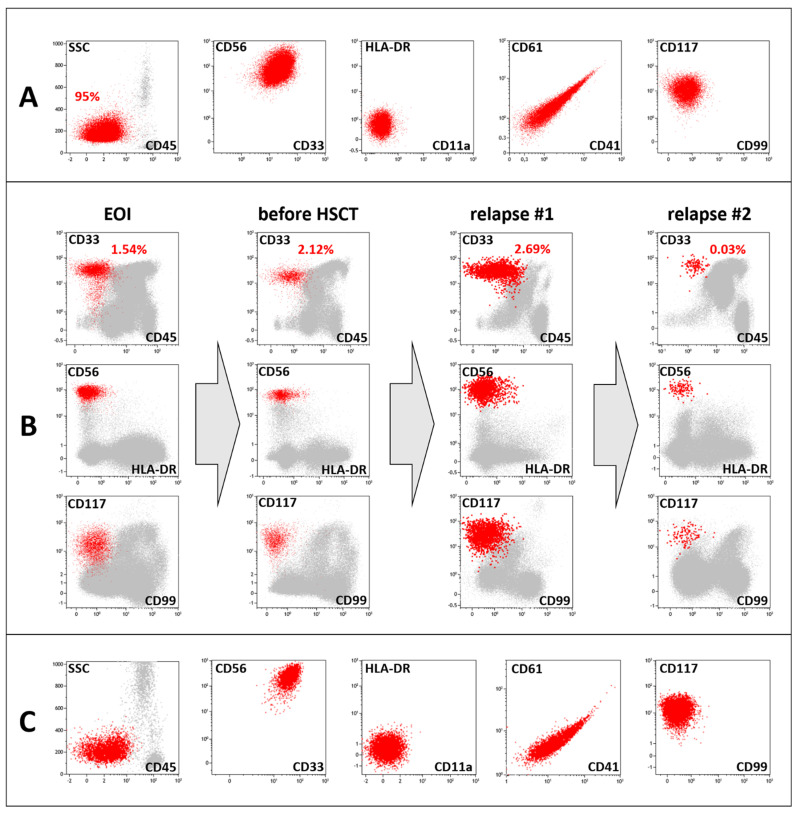
Flow cytometric studies in the presented case. (**A**)—The immunophenotype of BM blasts in primary AMKL; (**B**)—illustrative dot plots for the key points of MFC-MRD monitoring in the BM; (**C**)—the immunophenotype of leukemic blasts in the extramedullary mass biopsy at the second relapse. The leukemic blasts are red, whereas the remaining nucleated cells are gray.

**Figure 3 ijms-26-05895-f003:**
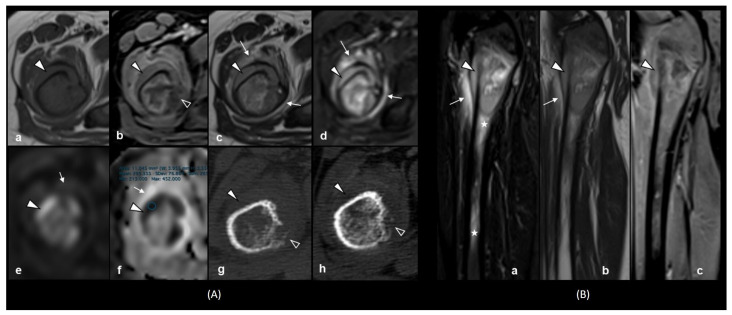
MR and CT images from the second relapse. Axial MR and CT images (**A**): T1-weighted (**a**), contrast-enhanced, fat-suppressed T1-weighted (**b**), T2-weighted (**c**), STIR (**d**), DWI (**e**) and an ADC map with an area with restricted diffusion (0.295 × 10^−3^ mm^2^/s) (**f**) sequences. Pre- (**g**) and postcontrast (**h**) CT scans demonstrating the subperiosteal extramedullary lesion (arrowheads), peripheral soft tissue edema (white arrows) and spiculated periosteal reaction with cortical destruction (empty arrowheads) at the proximal right femur. Sagittal MR images (**B**): STIR (**a**), T2-weighted (**b**) and contrast-enhanced T1-weighted (**c**) sequences revealed subperiosteal extramedullary relapse of AMKL (arrowheads) and intramedullary bone marrow edema (stars) extending to the middle of the diaphysis with peripheral soft-tissue edema (white arrow).

**Figure 4 ijms-26-05895-f004:**
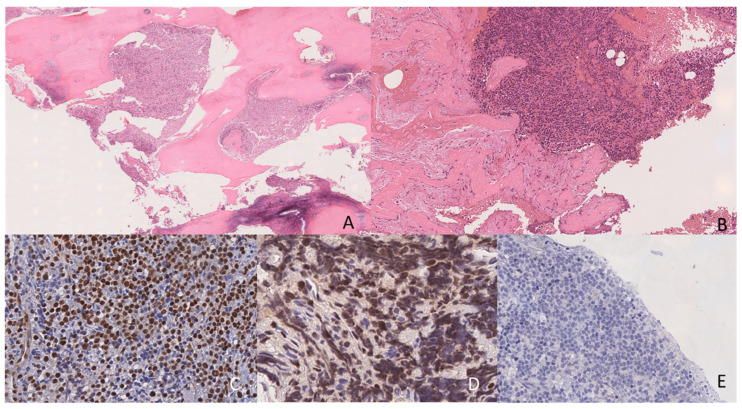
Histopathological and immunohistochemical findings. (**A**,**B**) Hematoxylin (blue) and eosin (shades of red) staining revealed a malignant small-, round-cell tumor with characteristic morphological features (×200 and ×400, respectively). Immunohistochemical staining revealed strong nuclear positivity for FLI-1 and ERG (dark brown, (**C**) and (**D**), respectively). At the same time, tumor cells were negative for CD61 (**E**). Nuclei are stained with hematoxylin (blue). Most other specific markers were nonreactive. Given the tumor location and morphology, Ewing sarcoma was initially considered in the differential diagnosis, but further workup confirmed AMKL with extramedullary involvement.

**Table 1 ijms-26-05895-t001:** Fluorochrome-conjugated antibodies used in diagnostic assessment of bone marrow samples from described patient. PB, Pacific Blue; KrO, Krome Orange; i, intracellular antigen.

Tube	FITC	PE	ECD	PC5.5	PC7	APC	APC-Alexa700	APC-Alexa750	PB	KrO
1	CD66b	CD19	CD56	CD117	CD33	CD34	CD14	CD45	CD7	CD3
2	iLZ	iMPO			iCD22	iCD79a		iCD3		CD45
3	CD61	CD11a	CD34	CD117	CD33	CD2	CD5	CD41a	CD235a	CD45
4	CD64	NG2	CD34	CD117	CD33	CD11c	CD4	CD303	CD13	CD45
5	CD38	CD371	CD34	CD117	CD33	CD99	CD123	CD45RA	HLA-DR	CD45
6	CD15	CD11a	CD34	CD117	CD33	CD56	CD14	CD11b	HLA-DR	CD45

## Data Availability

The raw data supporting the conclusions of this article will be made available by the authors on request.

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
