# Peer review of "Extramedullary Relapse of CBFA2T3::GLIS2-Positive Megakaryoblastic Leukemia Mimicking Secondary Ewing Sarcoma: An Exemplary Case for the Diagnostic Trap"

_ijms, 2025, doi:10.3390/ijms26125895_

Round 1
Reviewer 1 Report
Comments and Suggestions for Authors
The manuscript by Lebedeva et al provides a case report of an isolated extramedullary involvement of CBFA2T3::GLIS2-positive relapsed AMKL, which was mis-diagnosed as secondary Ewing sarcoma due to RAM-phenotype and morphology.
This study shows the importance of further diagnostics when two disease entities (in this cases AMKL and Ewing sarcoma) resemble. The presence of the CBFA2T3::GLIS2 fusion gene and absence of Ewing specific fusion genes (e.g.EWSR1 fusions) shifted the diagnose from secondary Ewing Sarcoma towards relapsed extramedulary AMKL.
Q: with the AMKL at first diagnosis in mind, faster diagnosis of the relapsed AMKL could have be obtained by screening for the CBFA2T3::GLIS2 fusion gene in the extramedulary biopsy and subsequent specific treatment. Could this have saved the patient, or at least delayed the girl's death? This would be interesting to adress in the discussion section
Author Response
Comments 1: with the AMKL at first diagnosis in mind, faster diagnosis of the relapsed AMKL could have be obtained by screening for the CBFA2T3::GLIS2 fusion gene in the extramedulary biopsy and subsequent specific treatment. Could this have saved the patient, or at least delayed the girl's death? This would be interesting to adress in the discussion section
Response 1: Thank you very much for pointing this out! Due to an administrative error of local clinical department, pathologists did not have access to the disease history. Indeed, screening for the CBFA2T3::GLIS2 fusion gene in the extramedulary biopsy material could reduce the time to diagnosis of the relapsed AMKL, but we suppose it would not significantly delay the patient death. We did not consider it necessary to go into these details in the text, because a similar diagnostic problem may also occur in primary patients, not only in relapsed AL.
Reviewer 2 Report
Comments and Suggestions for Authors
This case report highlights a rare and diagnostically challenging scenario of extramedullary relapse of CBFA2T3::GLIS2-positive AMKL, initially misdiagnosed as Ewing sarcoma. While the case is clinically significant, several critical issues must be addressed to improve clarity, methodological rigor, and scientific impact.
Major Issues:
- Timeline and Patient History Ambiguities
The patient’s age is inconsistently reported (1.7 years at initial diagnosis vs. 3.5 years at relapse). A clear timeline (e.g., a figure) is essential to track disease progression, treatments, and relapses. The abstract states the patient died “5 months after diagnosis,” but it is unclear whether this refers to the initial AMKL diagnosis or the extramedullary relapse. Clarification is critical for interpreting outcomes.
- Diagnostic Methodology Limitations
Insufficient Biopsy Material: The initial biopsy’s inadequacy for molecular confirmation is mentioned but not critically discussed. This procedural shortcoming should be analyzed as a potential contributor to misdiagnosis.
EWSR1 Exclusion: While EWSR1 rearrangements were ruled out, the report does not specify whether other sarcoma-associated fusions (e.g., FUS, CIC) or markers (e.g., NKX2.2) were tested. This weakens the differential diagnosis.
Imaging Analysis: ADC values are cited but not contextualized. How do these values compare to typical ranges for myeloid sarcoma vs. Ewing sarcoma? A deeper discussion of imaging’s role in differential diagnosis is needed.
- Methods Section Deficiencies
Antibody Panels: The “10-color combinations of monoclonal antibodies” lack specificity. The exact antibodies used for immunophenotyping (e.g., clones, vendors) must be listed for reproducibility.
RNA-seq Validation: The Arriba algorithm is mentioned, but validation steps (e.g., Sanger sequencing, orthogonal PCR) are absent. Without confirmation, the fusion’s presence remains uncertain.
FISH Probe Details: The commercial FISH probes (Cytocell, Kreatech) are named, but their validation in pediatric AMKL or Ewing sarcoma is not addressed. Were positive/negative controls included?
- Treatment Rationale and Outcomes
The choice of palliative chemotherapy (low-dose cytarabine + 6-mercaptopurine) post-relapse is not justified. Why were targeted therapies (e.g., GLIS2 inhibitors) or clinical trials not considered?
The high MRD persistence after HSCTs suggests treatment resistance. A discussion of potential mechanisms (e.g., immune evasion, fusion-driven pathways) is missing.
Minor Issues:
Writing and Formatting
Grammar/Syntax: Frequent semicolon misuse (e.g., “Fili1; ERG; and LMO2”) and inconsistent tense (e.g., shifts between past and present tense).
Abbreviations: Terms like MFC-MRD and HSCT are not defined at first use. Ensure all abbreviations are spelled out initially.
Figure and Table Accessibility
Figures are referenced but not fully present in the submission. Their absence hinders evaluation of key data (e.g., FISH images, flow cytometry plots). Ensure figures are provided with high resolution and annotations.
Discussion Limitations
The comparison of AMKL and Ewing sarcoma focuses on morphology and CD99 expression but neglects molecular contrasts (e.g., GLIS2-driven pathways vs. EWSR1-ETS fusions).The ADC values’ diagnostic utility is underdeveloped. Include recommendations for incorporating imaging biomarkers into clinical workflows.
References
Some citations (e.g., references 2-3) are outdated. Update with recent literature on AMKL genomics (e.g., 2023 WHO classification, newer prognostic studies).
Recommendations for Revision:
- Clarify Timeline: Add a table/figure summarizing the patient’s clinical course, including ages, treatments, and key events.
- Expand Methods:
- Specify antibodies, FISH probe validation, and RNA-seq confirmation steps.
- Detail exclusion of other sarcoma-related fusions/markers.
- Strengthen Discussion:
- Address mechanisms of treatment resistance and therapeutic implications of CBFA2T3::GLIS2.
- Propose a diagnostic algorithm for extramedullary AMKL vs. solid tumors.
- Ethical Compliance: Include IRB approval statement.
- Proofreading: Correct grammatical errors and standardize abbreviations.
This case underscores the diagnostic pitfalls in pediatric AMKL with extramedullary relapse. However, methodological gaps and insufficient clinical context diminish its impact. Addressing these issues will enhance the report’s utility for clinicians and researchers. Major revisions are required before publication.

The language should be improved.
Author Response
Comments 1: Timeline and Patient History Ambiguities: The patient’s age is inconsistently reported (1.7 years at initial diagnosis vs. 3.5 years at relapse). A clear timeline (e.g., a figure) is essential to track disease progression, treatments, and relapses. The abstract states the patient died “5 months after diagnosis,” but it is unclear whether this refers to the initial AMKL diagnosis or the extramedullary relapse. Clarification is critical for interpreting outcomes.
Response 1: Thank you very much for pointing this out! At the time of the initial diagnosis, the patient was 1.7 years old; at the time of diagnosis of the described relapse, the patient was 3.5 years old. 5 months after biopsy and diagnosis of extramedullary relapse, the patient died. We have corrected the text (highlighted in lines 23, 31, 105, 192) to clear this point.
Comments 2: Diagnostic Methodology Limitations: Insufficient Biopsy Material: The initial biopsy’s inadequacy for molecular confirmation is mentioned but not critically discussed. This procedural shortcoming should be analyzed as a potential contributor to misdiagnosis. EWSR1 Exclusion: While EWSR1 rearrangements were ruled out, the report does not specify whether other sarcoma-associated fusions (e.g., FUS, CIC) or markers (e.g., NKX2.2) were tested. This weakens the differential diagnosis. Imaging Analysis: ADC values are cited but not contextualized. How do these values compare to typical ranges for myeloid sarcoma vs. Ewing sarcoma? A deeper discussion of imaging’s role in differential diagnosis is needed.
Response 2:
Insufficient Biopsy Material: The material of the primary biopsy of bone lesion was admitted for the molecular tests for detection of Ewing-specific genetic rearrangements. Probably, due to intensive de-calcification of the material, the extracted RNA had high level of fragmentation (median fragment length was <90 bases as determined by TapeStation capillary electrophoresis instrument) and was not sufficient for molecular investigations neither by reverse-transcription PCR nor RNA sequencing.
EWSR1 Exclusion: Thank you very much for this comment. FUS, CIC fusions were excluded based on the results of RNAseq. We have added this information to the text (highlighted in lines 29, 187). Also immunohistochemical marker NKX2.2 was negative, we have added this information to the text (highlighted in lines 151-152).
Imaging Analysis: Thank you for pointing this out! We were pleased to have added a more detailed description of the obtained visualization data and the imaging’s role in differential diagnosis (highlighted in lines 108-128).
Comments 3: Methods Section Deficiencies: Antibody Panels: The “10-color combinations of monoclonal antibodies” lack specificity. The exact antibodies used for immunophenotyping (e.g., clones, vendors) must be listed for reproducibility. RNA-seq Validation: The Arriba algorithm is mentioned, but validation steps (e.g., Sanger sequencing, orthogonal PCR) are absent. Without confirmation, the fusion’s presence remains uncertain. FISH Probe Details: The commercial FISH probes (Cytocell, Kreatech) are named, but their validation in pediatric AMKL or Ewing sarcoma is not addressed. Were positive/negative controls included?
Response 3: Thank you very much for your thorough analysis!
Antibody Panels: We have expanded the methodological section in order to make the technology clear. Antibodies are listed in the Table 1.
RNA-seq Validation: CBFA2T3::GLIS2 fusion transcript was validated by Sanger sequencing, material was added to Figure 1.
FISH Probe Details: We have performed a validation of CBFA2T3::GLIS2 FISH probe on a large AML cohort of patients subjected to FISH and whole-genome sequencing. Convergence was found to be 100% (5 FISH-positive WGS-positive vs 278 FISH-negative WGS-negative). However, Ewing sarcoma FISH probes were not validated in the same setting due to unavailability of fresh tissue samples. We find that it is unnecessary to go into these details in the text, since the FISH probes used are commercial ones
Comments 4: Treatment Rationale and Outcomes: The choice of palliative chemotherapy (low-dose cytarabine + 6-mercaptopurine) post-relapse is not justified. Why were targeted therapies (e.g., GLIS2 inhibitors) or clinical trials not considered? The high MRD persistence after HSCTs suggests treatment resistance. A discussion of potential mechanisms (e.g., immune evasion, fusion-driven pathways) is missing.
Response 4: Thank you very much for your comment! Pediatric AMKL with the CBFA2T3::GLIS2 fusion is strongly associated with a poor prognosis and a high relapse incidence. We did not aim to discuss potential mechanisms of resistance development in this type of AMKL. Probably inclusion of such statements and discussion would enlarge the size of the paper greatly and blur the main message of the paper. The main point of this article is to highlight the difficulties of differential diagnosis between leukemia relapse and the development of a secondary tumor. The choice of palliative chemotherapy in the extramedullary relapse was made without the participation of the authors of this article and according to the available treatment options.
Comments 5: Writing and Formating: Grammar/Syntax: Frequent semicolon misuse (e.g., “Fili1; ERG; and LMO2”) and inconsistent tense (e.g., shifts between past and present tense). Abbreviations: Terms like MFC-MRD and HSCT are not defined at first use. Ensure all abbreviations are spelled out initially.
Response 5: Thank you very much for your remark! Frequent semicolon misuse (e.g., “Fili1; ERG; and LMO2”): We absolutely agree with your comment, however the semicolons in the original article are located differently, apparently, an automatic replacement occurred during formatting the file into PDF format. Inconsistent tense; abbreviations: Corrected.
Comments 6: Discussion Limitations: The comparison of AMKL and Ewing sarcoma focuses on morphology and CD99 expression but neglects molecular contrasts (e.g., GLIS2-driven pathways vs. EWSR1-ETS fusions).The ADC values’ diagnostic utility is underdeveloped. Include recommendations for incorporating imaging biomarkers into clinical workflows.
Response 6: Thank you for your remark! Conventional CT and MR imaging with DWI sequences and ADC map were performed as primary imaging. Control MRI for prognostic purposes, or for monitoring the response, to therapy was not carried out. Radionuclide diagnostic include radionuclide imaging of possible treatment targets and radionuclide therapy were not performed in our case. ADC values are a strong marker for determining tumor malignancy, which correlates with its high cellularity. Specific cut-off ADC values for differentiating myeloid sarcomas or Ewing's sarcoma are currently not defined. We hope that our case will contribute to further study of imaging parameters of extramedullary relapses and myeloid sarcomas in a pediatric cohort. Discussions on ADC values have been added to the "discussion" section (highlighted in lines 246-266).
Comments 7: References: Some citations (e.g., references 2-3) are outdated. Update with recent literature on AMKL genomics (e.g., 2023 WHO classification, newer prognostic studies).
Response 7: Thank you very much for your remark! References 2-3 represent «classic» description of the M7 “morphological” variant of AML, so we suppose that they should not be considered outdated. Nevertheless, we have added a new reference to the 5th edition of the WHO classification.
Round 2
Reviewer 2 Report
Comments and Suggestions for Authors
Review Report
Manuscript Title: Extramedullary relapse of CBFA2T3::GLIS2-positive megakaryoblastic leukemia mimicking secondary Ewing sarcoma: an exemplary case for the diagnostic trap
This manuscript presents a compelling and rare pediatric case of CBFA2T3::GLIS2-positive acute megakaryoblastic leukemia (AMKL) with isolated extramedullary relapse mimicking Ewing sarcoma. The clinical and pathological overlap is discussed thoroughly, emphasizing the diagnostic pitfalls. This is a valuable case with real-world relevance. The authors also provided detailed diagnostic workup with integration of morphology, immunophenotyping, FISH, and RNA-seq. However, there’are still some concerns maybe the authors need to clarify.
- Diagnostic Ambiguity: The diagnosis of relapse is sometimes asserted too onfidently before definitive confirmation. Terms like “suspected relapse” or “presumptive” should be used until FISH/RNA-seq results are available.
- Imaging Interpretation: ADC values are used conclusively, but caution is warranted due to variability in thresholds and lack of standardized cutoffs for pediatric hematologic malignancies.
- Therapeutic Justification: Key treatment decisions, such as second HSCT or use of venetoclax/decitabine, are not well substantiated with literature support.
- Clarify the immunophenotypic overlap between AMKL and Ewing sarcoma, Improve figure legends (e.g., explain what FLI-1 positivity implies in this context).Consider revising the title and abstract for brevity and clarity.
- Writing Quality: The manuscript contains multiple grammatical, typographical, and structural issues that detract from its readability and professionalism.
The manuscript would benefit from a professional English-language editing service. Authors should expand on treatment rationale, clarify MRD methodology, and improve figure/table presentation.
Author Response
Comments 1: Diagnostic Ambiguity: The diagnosis of relapse is sometimes asserted too сonfidently before definitive confirmation. Terms like “suspected relapse” or “presumptive” should be used until FISH/RNA-seq results are available.
Response 1: Thank you very much for your comment! We've changed the title of the third section to make the article easier to understand.
Comments 2: Imaging Interpretation: ADC values are used conclusively, but caution is warranted due to variability in thresholds and lack of standardized cutoffs for pediatric hematologic malignancies.
Response 2: Thank you very much for pointing this out! We added a clarifying comment in the Discussion section.
Comments 3: Therapeutic Justification: Key treatment decisions, such as second HSCT or use of venetoclax/decitabine, are not well substantiated with literature support.
Response 3: Unfortunately, we cannot discuss the process of making therapy decisions, especially after the first relapse, because they were made without our participation. Moreover, in such adverse cases, therapy decisions after the failure of first-line treatment are completely personalized; therefore, we cannot comment on them. We can only tell the patient's story. On the other hand, we must say that our manuscript is not devoted to the choice of second-line treatment in relapsed high-risk AML. We wanted to focus the readers' attention on the diagnostic problem, which can also occur during initial diagnostics
Comments 4: Clarify the immunophenotypic overlap between AMKL and Ewing sarcoma, Improve figure legends (e.g., explain what FLI-1 positivity implies in this context). Consider revising the title and abstract for brevity and clarity.
Response 4: We tried to explain the difficulties in the immunohistochemistry data interpretation as well as the role of FLI-1 in the doubts of the final diagnosis. The abstract was shortened to make our message clear.
Comments 5: Writing Quality: The manuscript contains multiple grammatical, typographical, and structural issues that detract from its readability and professionalism.
Response 5: Thank you very much for your comment! We reviewed the text and corrected the errors. Additionally, we enclose a certificate of English language verification from the Springer Nature.